# Analysis of Volatile Organic Compounds in Exhaled Breath Following a COMEX-30 Treatment Table

**DOI:** 10.3390/metabo13030316

**Published:** 2023-02-21

**Authors:** Feiko J. M. de Jong, Thijs T. Wingelaar, Paul Brinkman, Pieter-Jan A. M. van Ooij, Anke H. Maitland-van der Zee, Markus W. Hollmann, Rob A. van Hulst

**Affiliations:** 1Royal Netherlands Navy Diving and Submarine Medical Centre, 1780 CA Den Helder, The Netherlands; 2Department of Anesthesiology, Amsterdam UMC, Location AMC, 1100 DD Amsterdam, The Netherlands; 3Department of Pulmonology, Amsterdam UMC, Location AMC, 1100 DD Amsterdam, The Netherlands

**Keywords:** hyperbaric oxygen therapy, hyperoxia, pulmonary oxygen toxicity, exhaled breath markers, volatile organic compounds, COMEX-30, GC-MS, diving and hyperbaric medicine

## Abstract

The COMEX-30 hyperbaric treatment table is used to manage decompression sickness in divers but may result in pulmonary oxygen toxicity (POT). Volatile organic compounds (VOCs) in exhaled breath are early markers of hyperoxic stress that may be linked to POT. The present study assessed whether VOCs following COMEX-30 treatment are early markers of hyperoxic stress and/or POT in ten healthy, nonsmoking volunteers. Because more oxygen is inhaled during COMEX-30 treatment than with other treatment tables, this study hypothesized that VOCs exhaled following COMEX-30 treatment are indicators of POT. Breath samples were collected before and 0.5, 2, and 4 h after COMEX-30 treatment. All subjects were followed-up for signs of POT or other symptoms. Nine compounds were identified, with four (nonanal, decanal, ethyl acetate, and tridecane) increasing 33–500% in intensity from before to after COMEX-30 treatment. Seven subjects reported pulmonary symptoms, five reported out-of-proportion tiredness and transient ear fullness, and four had signs of mild dehydration. All VOCs identified following COMEX-30 treatment have been associated with inflammatory responses or pulmonary diseases, such as asthma or lung cancer. Because most subjects reported transient pulmonary symptoms reflecting early-stage POT, the identified VOCs are likely markers of POT, not just hyperbaric hyperoxic exposure.

## 1. Introduction

In a hyperbaric environment, such as while scuba diving or working in caisson construction or hyperbaric chambers, inert gas molecules such as nitrogen dissolve in tissue due to increased ambient pressure. When ambient pressure decreases after hyperbaric exposure, dissolved gas must be exhaled gradually to prevent the formation of bubbles in tissue or bodily fluids.

Decompression sickness (DCS) is an acute illness caused by the saturation of inert gases in bodily tissues and/or fluids after a decrease in ambient pressure and can cause a variety of symptoms, ranging from negligible sensations to severe neurological deficit or even death [1,2]. Because a substantial decrease in ambient pressure is required, professional and recreational divers are most often affected, although airplane crews, hyperbaric chamber inside tenders, and caisson workers may also be at risk [3,4]. Whereas the preclinical treatment for DCS consists of normobaric oxygen treatment and rehydration, its definitive management consists of recompression therapy using hyperbaric treatment tables with 100% oxygen or a hyperoxic gas mix [5,6].

The COMEX-30 hyperbaric treatment table is used by the Diving and Submarine Medical Centre of the Royal Netherlands Navy to treat patients with severe dysbaric illnesses, such as spinal or inner ear decompression sickness [7]. Unlike the shorter oxygen tables commonly used by the US Navy, such as treatment tables 5 and 6, which start at a pressure equal to 18 m of seawater (msw), the treatment depth of a COMEX-30 table starts at a pressure equal to 30 msw. Patients who undergo treatment with a COMEX-30 table inhale a gas mixture of 50% helium and 50% oxygen to mitigate the risk of oxygen toxicity, an adverse effect of breathing a gas mix with an increased partial pressure of oxygen (*p*O_2_). Oxygen toxicity has been characterized as central nervous system oxygen toxicity (CNS-OT) or pulmonary oxygen toxicity (POT), depending on the *p*O_2_ and the duration of hyperoxic exposure [8]. The initial symptoms of CNS-OT have been reported at a *p*O_2_ of 1.3–1.6 atmosphere absolute (ATA), whereas the first symptoms of POT have been found to occur at 0.5 ATA [9].

POT is believed to be caused by reactive oxygen species that trigger cellular damage and inflammatory responses, mainly in the lung epithelium and alveoli [10]. Symptoms of POT include retrosternal discomfort, coughing, end-inspiratory burning sensation, and shortness of breath and are caused by inflammatory responses and interstitial pulmonary edema, called the exudative phase. Although these symptoms of POT are fully reversible when *p*O_2_ is decreased, prolonged exposure can lead to irreversible lung damage, such as pulmonary fibrosis and lung function impairment, also known as the proliferative phase [11,12,13].

Various methods to predict and detect POT have been investigated. For example, the Unit of Pulmonary Toxicity Dose (UPTD) is based on a decrease in vital capacity after hyperbaric and hyperoxic exposures [14,15], with 1 UPTD being equal to breathing 100% oxygen at 1 ATA for 1 min. Although this scale was developed with dry hyperbaric exposures with subjects at rest, it remains frequently used to predict the risk of POT during both dry/chamber and wet/in-water diving. At present, 615 UPTD is considered the daily safe limit for dry hyperbaric exposure, corresponding to a 2% decrease in vital capacity in 50% of the population, with a maximum limit of 1425 UPTD for exceptional cases [16].

Other models have been developed since the UPTD. Arieli introduced the POT-index or K-index, which allows for a more precise estimation of POT and can provide an estimation of POT incidence and recovery time needed after hyperoxic exposure [17]. As with the UPTD model, the K-index model is based on the vital capacity derived from pulmonary function tests and has its limitations, especially when calculating repetitive dives [12].

The Royal Netherlands Navy has evaluated several methods for detecting POT after hyperbaric hyperoxic exposure, with molecular breath analysis by gas chromatography-mass spectrometry (GC-MS) being the most promising [18]. Subsequent analysis of exhaled breath by GC-MS included electronic nose detection of POT [19]. A recent analysis of molecules associated with POT in the breath of subjects after a treatment table 6 reported a decrease in molecules and molecule classes similar to those reported in earlier studies by Van Ooij and Wingelaar [18,19].

All of these earlier studies applied hyperbaric exposures under 615 UPTD, except for the treatment table 6 study with 641 UPTD [20], with none of these subjects showing any clinical signs of POT. Therefore, molecules identified in those studies were markers of pulmonary hyperoxia, not POT. Because oxygen exposure of a COMEX-30 treatment table corresponds to 1045 UPTD, the present study hypothesized that VOCs identified following a COMEX-30 would be associated with POT.

## 2. Materials and Methods

### 2.1. Preparation and Hyperbaric Exposure

The sampling protocol was approved by the Medical Ethical Committee of the Amsterdam Academic Medical Center for breath sample collection (ref. W18_424 # 21.083). The hyperbaric exposure was part of the regular hyperbaric training of hyperbaric chamber operators and medical staff; thus, prior ethical approval was not required. However, all procedures were performed in accordance with the declaration of Helsinki, the ICH GCP E6(R2) Good Clinical Practice guidelines, and with the authorization of the Surgeon General of the Netherlands Ministry of Defence [21]. All procedures adhered to the precautionary measures to guarantee the safety of the subjects, as defined by standard chamber operation instructions. Participation was voluntary; all subjects provided written informed consent and could withdraw from the study at any time without providing an explanation. In compliance with national privacy legislation and European Data Protection Regulations (GDPR), no data obtained during this study were included in the participants’ medical files.

All participants were healthy, nonsmoking volunteers, consisting of experienced divers or hyperbaric nurses in the Netherlands navy. All were medically fit for hyperbaric exposure in accordance with the medical fitness requirements for diving of the Netherlands Ministry of Defence and the Dutch national policy on medical standards for occupational diving [22]. Exclusion criteria were recent respiratory tract infections and consuming an average of two alcoholic beverages per day. It was not allowed to have strenuous physical activities or consume alcohol on the day before the study. Similarly, hyperbaric exposure of any sort was not allowed within 72 h before the day of the study. On the day of the study, all subjects were required to report current physical symptoms and all medications or supplements that they were taking, even over-the-counter products, with final inclusion determined on an individual basis to minimize contamination of breath samples.

All participants consumed a uniform diet, consisting of bread and marmalade, a low-lipid nutritional drink, and water, on the day of exposure to limit contamination of the breath samples and minimize confounders. To prevent metabolic alterations due to fasting, the subjects were encouraged to eat and drink during the day and did so freely, including during the hyperbaric exposure, but intake of any substance except for water was not allowed within 1 h before the measurements to prevent contamination of the breath samples.

The hyperbaric exposure was performed in the multiplace recompression chamber (Haux Life Support, Karlsbad, Germany) of the Diving and Submarine Medical Centre of the Royal Netherlands Navy. The exposure was equal to a COMEX-30 treatment table. During this treatment table, the subjects were recompressed to 4 ATA (405 kPa; equivalent to a depth of 30 msw) and started breathing a mixture of 50% helium and 50% oxygen, followed by 100% oxygen when a depth of 2.8 ATA (283 kPa; 18 msw) was reached. According to the original COMEX-30 schedule, at regular intervals an air break was included with the subjects breathing ambient (chamber) air for 5 min to lower the risk of oxygen toxicity. As all subjects were experienced divers or hyperbaric nurses, no inside tender was present during the study. In case of an emergency, additional staff could enter the chamber. The entire hyperbaric exposure lasted 7 h 30 min (see Appendix A for a graphic representation of the treatment table).

### 2.2. Collection and GC-MS Analysis of Breath Samples

The sampling protocol was identical to that reported in previous studies [20,23,24,25]. Four breath samples per subject were collected: one before hyperbaric exposure and three after exposure, including one each at 30 min, 2 h, and 4 h after “surfacing.” At each measurement, the subjects had to breathe for 5 min through an inspiratory VOC filter (Honeywell, Charlotte, NC, USA) fitted to a one-way valve. A nonelastic sampling balloon (Globos Nordic, Naestved, Denmark) was attached to the one-way valve exhaust, and a large exhalation was blown into the balloon. The balloon was sealed, and a gas pump (Gastec, Kanagawa, Japan) was used to pump 500 mL of exhaled air through a sampling tube (Tenax GR 60/80, Camsco, Houston, TX, USA) in 2 min. To identify possible contaminants from the test environment, various samples from the surroundings were collected, including ambient air from the interior of the recompression chamber, the breathing masks, the sampling area, and the diet.

After the last breath samples were collected, the subjects were questioned about their experience of any enduring signs of POT and other symptoms. This inquiry was repeated 24 h after completion of the hyperbaric exposure.

The breath samples were processed using the TD-GC-MS setup (GC-MS QP2010; Shimadzu, Japan + TD100; Markes, Sacramento, CA, USA) of the University of Amsterdam. Briefly, the sampling tubes were heated in the thermal desorption unit at 250 °C for 15 min at a flow rate of 30 mL/min, followed by a 10 °C cold trap and reheating to 300 °C for 1 min. The VOCs were splitlessly injected into a 30 m gas chromatography column (Restec, Bellefonte, PA, USA), 0.25 mm in diameter, at a rate of 12 mL/min. The molecules were electronically ionized at 70 eV and captured using a quadruple spectrometer with a range of 37–300 Da.

### 2.3. Statistical Analysis and Identification of Compounds

An earlier study of exhaled breath markers after a treatment table 6 reported variations in VOCs of up to 45% [20]. Because the hyperoxic exposure in the present study was longer, and the subjects were exposed to a higher pressure, similar or greater variations were expected. Assuming a power of 80% and a significance level of 0.05, a minimum of nine subjects would be required to detect such changes and reject the null hypothesis.

The GC-MS output was processed for peak detection, denoising, and retention time alignment, as described previously [8,20,23,24,25]. Significant differences in ion fragments per retention time were compared between the baseline measurement and the measurements after hyperbaric exposure by Wilcoxon signed-rank tests. If univariate analysis identified three or more ion fragments (hereafter called “fragment cluster”) at a certain retention time, the chromatograms with the most distinctive peaks at the corresponding time were examined, with the VOC identified by a similarity search in the NIST webbook library (National Institute of Standards and Technology, National Bureau of Standards; US Department of Commerce, Gaithersburg, MD, USA).

PubChem [26], the Human Metabolome Database (HMDB) [27], the Chemical Entities of Biological Interest database (ChEBI) [28], and Pubmed [29] were consulted to evaluate the associations of the identified molecules with the human endogenous metabolome, including various pulmonary diseases, presence in cell membrane structures, and associations with cellular inflammatory responses. If the identified molecule was highly abundant in environmental samples, or frequently reported to be exogenous, that compound was regarded as a contaminant. If the similarity search identified multiple viable compounds with similar percentages of resemblance, such as isomers, and the online library analysis did not unambiguously identify the unknown compound, the remaining uncertainty was resolved by discussions among three authors (F.J.M.d.J., T.T.W., and P.B.) until a consensus was reached.

Because one sample was missing, the Skillings-Mack test rather than the Friedman test was employed to determine the overall significance of signal intensities of the VOCs over various measurements [30]. If a significant variability was found, a post hoc analysis was performed using the Wilcoxon signed-rank test with Bonferroni correction. Figure 1 provides an overview of the sampling, statistical analysis, and identification of the data.

All data were compiled and statistically processed using R Statistical Software (v4.0.3; R Core Team 2021), combined with R-packages MBESS (v4.9.0; Kelley 2022), SVA (v3.20.0; Leek and Storey 2008), Rstatix (v0.7.0; Kassambara 2021), and XCMS (v3.12.0; Smith et al., 2006). All chromatograms were examined using GC-MS Postrun Analysis software (GC-MS Solution version 4.52, Shimadzu Corporation). An alpha of 0.05 was considered statistically significant. 

## 3. Results

Eleven healthy volunteers participated in the trial; one was excluded on the day of hyperbaric exposure due to a respiratory tract infection. The ten participants consisted of seven men and three women, of median age 35 years (range 31–38 years) and with a mean body mass index (BMI) of 25 ± 2 kg/m^2^ (Table 1). There were no deviations from protocol during hyperbaric exposure, sampling, or periods between sampling.

### 3.1. Symptoms Experienced after Hyperbaric Exposure

All subjects experienced some minor symptoms after hyperbaric exposure (Table 2). Seven subjects reported mild pulmonary symptoms associated with POT, ranging from an itching perception in the lower airways to an end-inspiratory retrosternal burning sensation. Most of these symptoms were experienced 30 min and 2 h after the end of exposure, and to a lesser degree at 4 h. Six subjects experienced constitutional symptoms, such as fatigue or the sensation of being dehydrated. Fatigue was reported by five subjects, being described as out-of-proportion tiredness in the evening or the day following hyperbaric exposure, comparable to having a bad night’s sleep. Four subjects reported symptoms of mild dehydration the day after the trial, manifested as concentrated urine and feeling unusually thirsty during the night after the exposure. Six persons reported upper airway symptoms; three subjects experienced dry or irritated nasal mucosa, whereas five subjects had a transient sensation of fullness of the middle ears, called “oxygen ear” by divers, on the morning after the trial, but this resolved within a few hours [31].

### 3.2. Identified Compounds

A total of 48 sample tubes were collected, including eight environmental samples. One sample from a test subject was excluded due to an error in the sampling procedure. After denoising, automatic peak detection, and alignment, 3104 ion fragments were observed in the remaining 39 samples. Univariate testing for significant variations between pre- and postexposure measurements resulted in 315 fragments, constituting 16 fragment clusters, within a retention time of 20 min. After further analysis, seven clusters were discarded due to the absence of a peak in the corresponding chromatograms or a lack of human associations with the identified molecules. Most excluded compounds were also detected in the environmental samples.

The nine remaining fragment clusters were identified as compounds associated with damage to cellular membranes, pulmonary pathology, or inflammatory responses. Four compounds (ethyl acetate, tridecane, nonanal, and decanal) showed significant variation in signal intensities after hyperbaric exposure (Appendix B). Ethyl acetate intensities were 5-fold higher in the samples obtained 2 and 4 h after exposure than in the pretreatment samples, whereas tridecane, nonanal, and decane intensities were 43%, 50%, and 33% higher, respectively, in the 2 or 4 h samples. Post hoc analysis demonstrated that the increase in ethyl acetate from baseline to 2 h was statistically significant (*p* = 0.023). The other five VOCs (2,4-dimethylpentane, methylcyclohexane, 3-methyleneheptane, octane, and butyl acetate) did not show significant variations over time (Appendix C).

Breath sample analysis of the seven subjects reporting pulmonary symptoms identified eight human-associated VOCs. Two additional compounds (3-methylheptane and 3-methylnonane) were identified, along with six VOCs that were previously found in the complete sampling cohort. (Appendix D). However, none of these eight compounds demonstrated significant differences in signal intensities. No relevant VOCs were present in the three subjects without pulmonary symptoms.

## 4. Discussion

An earlier study of breath analysis after a treatment table 6 in a recompression chamber identified 11 human-associated compounds [20]. Five of these molecules (3-methyleneheptane, octane, methylcyclohexane, nonanal, and dodecane) were also found in this study, although not all intensities varied significantly between baseline and postexposure measurements. Methylcyclohexane, 3-methyleneheptane, nonanal, and decanal were also previously detected after various types of hyperbaric hyperoxic exposure [23,24]. All of these studies, including the present study, found that the types of compounds that increased included mostly alkanes, alkenes, and aldehydes, indicating that similar molecules and classes of molecules are exhaled regardless of the type of hyperbaric hyperoxic exposure. These findings indicate that POT gradually develops from a preclinical phase after mild hyperbaric hyperoxic exposure to a symptomatic phase after more strenuous exposure. Although those studies reported no direct link with POT, as all the test subjects were asymptomatic after exposure, seven of the ten subjects exposed to a COMEX-30 table in the present study reported symptoms of POT, such as pulmonary discomfort and end-inspiratory burning sensations. These symptoms were mostly experienced at 30 min and 2 h after exposure, and to a lesser degree at 4 h. Thus, the identified VOCs might indicate early-stage POT, not just hyperbaric hyperoxic exposure.

Of the seven men, four reported pulmonary symptoms, and three reported constitutional symptoms; by contrast, all three women reported these symptoms. Moreover, most of the male reported upper airway symptoms such as “oxygen ear” or dry mucosa, compared with just one woman. Although these differences could be due to the small sample sizes, they may also result from differences in anatomy and physiology between men and women [32,33,34].

Although all participants were encouraged to drink during the day of the study including during the hyperbaric exposure, four subjects reported symptoms of mild dehydration during follow-up. This might be the result of breathing dry gases, but could also be the consequence of a reduced oral intake caused by reluctance to use the rudimentary sanitary facility in the hyperbaric chamber. The effect of this dehydration on VOC production remains unknown. Future studies should involve fluid intake- and output monitoring or bioimpedance measurements to quantify hydration status. Earlier studies of shorter hyperbaric exposure found that the signal intensities of most of the exhaled molecules tended to decrease [20,24]. In the present study, however, all molecular intensities increased after hyperoxic exposure, although most VOCs showed a slight, but not significant, decrease in intensity at 4 h. Increases in signal intensity have also been observed after in-water hyperoxic dives, which are thought to be more demanding of the lungs than dry hyperbaric exposure [23,25]. This finding suggests that the COMEX-30 table, like in-water dives, is more stressful for the lungs than other types of therapeutic recompression. 

Worldwide, short oxygen tables such as the US Navy treatment tables 5 and 6, are used more frequently than deeper tables with nitrox-oxygen or helium–oxygen gas mixtures, such as treatment table 6A and the COMEX-30 table [35,36]. Reports have suggested that the COMEX-30 table is more advantageous than the US Navy short oxygen tables in managing severe DCS, such as spinal cord DCS [37,38,39,40]. To date, however, no large-scale trials or reviews have assessed whether use of these deep tables is a standard of care for patients with severe DCS [36,41]. However, animal models have shown that breathing a helium–oxygen gas mixture after ischemic events has a neuroprotective effect, although most of these studies focused on brain ischemia, not on spinal cord lesions [42,43,44] or inner ear DCS [45]. Because short and deep oxygen tables yield comparable therapeutic outcomes, and use of the COMEX-30 table causes mild POT in healthy subjects even without hyperoxic exposure during the previous days, the shorter tables might be used. Helium–oxygen gas mixtures may, however, play an as yet undetermined role in the future of hyperbaric medicine.

### 4.1. Identified Compounds

Although all nine human-associated VOCs increased in intensity over time, only four showed significant changes, including the three compounds with the highest molecular weights (nonanal, decanal, and dodecane). Two of these compounds, nonanal and decanal, are aldehydes, with both found to be associated with pulmonary diseases such as lung cancer or asthma [46,47,48,49]. Aldehydes are formed in the body after tobacco exposure, during the metabolization of alcohols, and as a secondary product of lipid peroxidation [50,51]. Because smoking was a criterion for subject exclusion, and alcoholic beverages were prohibited on the day before the trial, exposure to tobacco and alcohol could be ruled out as major sources of these aldehydes. Alcohols in the body derive from ingestion of alcoholic beverages and from alkane metabolization by cytochrome p450 in the liver. Alkane levels in exhaled breath tend to rise after hyperbaric exposure, possibly contributing to the formation of aldehydes [50]. However, the metabolization of peroxidation damage is the most likely source of these aldehydes. In this process, Ω3 and Ω6 fatty acids in cell membranes are destroyed by exposure to reactive oxygen species, forming hydroperoxides, which are metabolized by cytochrome p450 to aldehydes [47,48,50].

Ethyl acetate is an ester found in the cytoplasm of all eukaryotes, including humans, as well as in various types of food as a naturally occurring compound or as part of an additive. In chemistry, it is used as a solvent [52]. Ethyl acetate was also detected in environmental samples, including the foods and drinks ingested by the test subjects. An exploratory multivariate PLS-DA analysis showed that ethyl acetate concentrations did not differ significantly at sampling times, indicating that other compounds might be more useful in detecting POT [53]. By contrast, the present study found that ethyl acetate concentrations were 5-fold higher 2 and 4 h after hyperbaric exposure than at baseline, whereas the other compounds increased only 33–50%. These findings did not correspond with food intake on the day of the trial as the subjects were allowed to eat during the last few hours of the hyperbaric exposure and did so freely. Thus, exposure to ethyl acetate in food started hours prior to the low intensity found at the 30 min measurement. Ethyl acetate has also been detected in breath samples of cancer patients and asthmatics [54,55,56].

Tridecane is a straight-chain alkane that has been associated with pulmonary diseases, such as asthma, lung cancer, and other malignancies [57,58]. These alkanes are generally believed to originate from hyperoxic damage by reactive oxygen species. During this process, called lipid peroxidation, the polyunsaturated fatty acids in the bilayer of cell membranes are destroyed, causing cellular damage and apoptosis and resulting in POT [10,59,60].

Analysis of the subjects with pulmonary symptoms identified two additional VOCs, 3-methylheptane and 3-methylnonane. Both of these VOCs are methylated alkanes previously identified after hyperbaric hyperoxic exposure [23,25,61]. In those studies, in which subjects were exposed to lower amounts of oxygen than in the present study, none of these participants reported pulmonary symptoms. This could indicate that preclinical exhalation of POT-associated VOCs increases with increasing oxygen exposure.

### 4.2. Strength and Limitations

To our knowledge, the present study is the first to analyze VOCs in subjects after heliox-oxygen hyperbaric exposure. Previous studies evaluated subjects breathing hyperoxic gas mixtures with nitrogen as a diluent or 100% oxygen [20,23,24,25,62]. Moreover, to our knowledge, earlier studies did not assess VOCs after such a long period of hyperbaric exposure, with equally high UPTDs, nor have subjects in any previous studies of VOCs reported initial symptoms of POT.

The present study, however, had several limitations. As with all laboratory studies in controlled environments and carefully selected test subjects, caution should be exercised in applying these results to real-life scenarios. For example, all subjects in this study were fit and healthy nonsmokers and were not exposed to pulmonary irritants, such as diving, normobaric oxygen, or upper airway infections, in the days before hyperbaric exposure. By contrast, a stricken diver treated with a COMEX-30 must be exposed to increased *p*O_2_ before the hyperbaric intervention: the causative dive with increased *p*O_2_ during the hours before treatment is a stressor of the airways, as is the inspired normobaric 100% oxygen during transfer to the hospital or undergoing examination in the emergency department. Thus, the lungs of a diver would have an even higher oxygen burden after COMEX-30 treatment than the subjects in this study, resulting in higher VOC signal intensities and corresponding pulmonary symptoms. Unfortunately, few studies of subjects treated with the COMEX-30 have reported rates of pulmonary discomfort and other adverse effects.

Another limitation of the present study was the lack of a non-pressurized control group for assessment of the experienced symptoms. We applied the self-controlled case series design, which is suitable for the longitudinal evaluation of VOC intensities. Blinding of subjects to hyperbaric exposure is difficult, especially as these subjects could sense whether the hyperbaric chamber is or is not pressurized [63]. Blinding was even more difficult in the present study as the test subjects were all extensively familiar with the hyperbaric chamber. These subjects probably would have noticed if the chamber was pressurized to a shallower depth than 30 msw, just by the density of the chamber atmosphere.

As reported above, we found a notable difference in the incidence of symptoms of POT between men and women; half of the male subjects reported lower airway symptoms, compared to all of the female subjects. While large studies have not shown differences in the exhaled breath profiles between males and females, a possible influence could not be determined due to the sample size not being large enough [64].

Finally, VOC identification is cumbersome, and the results are subject to multiple interpretations or biases such as availability and confirmation bias. The GC-MS identification software provides similarity scores of the targeted ion clusters when compared with the entire known molecular spectrum. If these scores are far apart, the identification of the correct molecule is relatively clear. However, when the similarity search yields comparable scores, and the suggested molecules are all human metabolites or cell components, identification becomes more challenging and can lead to an availability or confirmation bias. This is often encountered with isomers and other closely related molecules. Thus, if the similarity scores are alike, selection of the proper molecule must be accomplished by manually cross-referencing molecules in the literature and selecting the appropriate molecule as objectively as possible. To facilitate this process and reduce selection bias, a library has been created, consisting of VOCs frequently found after hyperbaric hyperoxic exposure [61]. Although this makes the identification process less prone to cherry-picking, greater input is necessary to further expand and validate this library.

## 5. Conclusions

As in earlier studies on VOCs after hyperbaric hyperoxic exposure, the four identified human-associated compounds consisted of alkanes, aldehydes, and esters. All are associated with inflammatory responses or with pulmonary diseases such as asthma or lung cancer. Because most of the included subjects reported transient symptoms, similar to those encountered in patients with early-stage POT, the VOCs identified in the present study may be markers for early-stage POT, not just hyperbaric hyperoxic exposure. Additional studies are needed to streamline the process of identifying and selecting exhaled compounds after hyperbaric hyperoxic exposure, to validate the identified compounds by analyzing breath samples from subjects with dives high in *p*O_2_, and ultimately to develop a robust method for breath analysis in the field during diving operations and in hyperbaric oxygen clinics to check for insipient POT.

## Figures and Tables

**Figure 1 metabolites-13-00316-f001:**
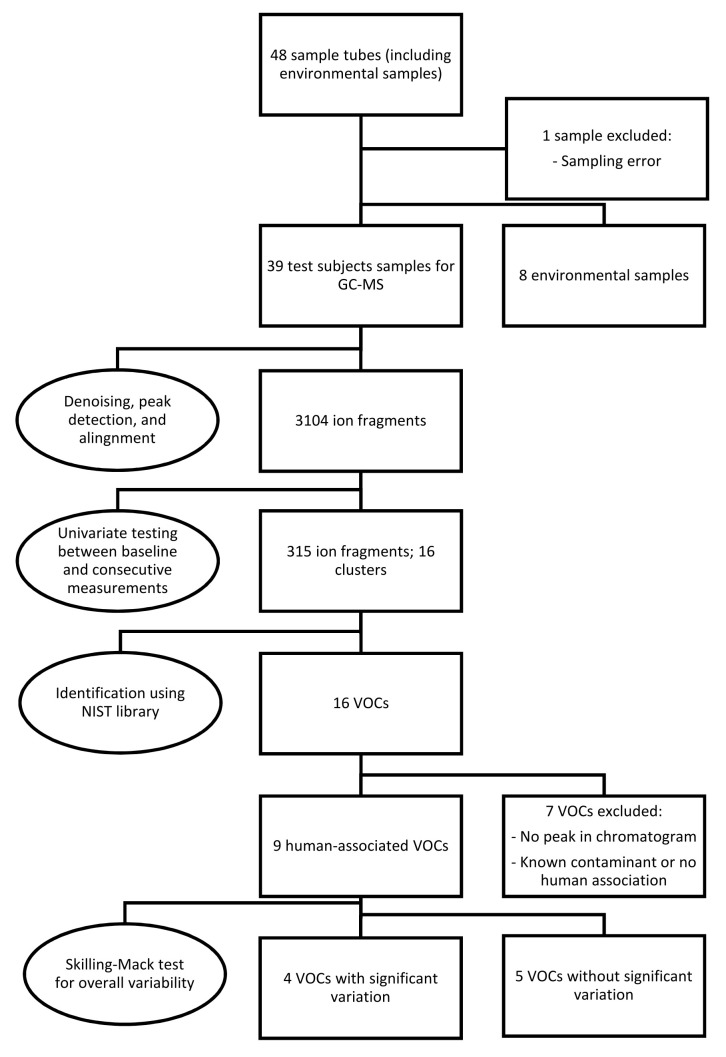
Overview of the methodology used in the present study, including breath sampling, statistical analysis, identification and selection of the data. Abbreviations: GC-MS, gas chromatography-mass spectrometry; NIST, National Institute of Science and Technology; VOCs, volatile organic compounds.

**Table 1 metabolites-13-00316-t001:** Baseline characteristics of final participants.

	Total (*n* = 10)	Men (*n* = 7)	Women (*n* = 3)
Age (years)	34.5 (7.8)	34 (4.5)	41 (12)
Height (cm)	180 (8.5)	184 (9)	171 (7)
Weight (kg)	79.5 (14.3)	85 (12)	69 (5)
BMI (kg/m^2^)	24.5 (1.3)	24.7 (1.4)	22.1 (0.8)

All results are reported as median (IQR). Abbreviation: BMI, body mass index.

**Table 2 metabolites-13-00316-t002:** Symptoms experienced by subjects after hyperbaric exposure.

	Men (*n* = 7)	Women (*n* = 3)	Total (*n* = 10)
Pulmonary/lower respiratory tract	4	3	7
Constitutional	3	3	6
*Fatigue*	2	3	5
*Dehydration*	2	2	4
Upper respiratory tract	5	1	6
*“Oxygen ear”*	4	1	5
*Dry mucosa*	2	1	3

## Data Availability

The data presented in this study are available on request from the corresponding author. The data are not publicly available because they belong to the Netherlands Ministry of Defence, and therefore cannot be shared unconditionally.

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
