# Peer review of "Analysis of Volatile Organic Compounds in Exhaled Breath Following a COMEX-30 Treatment Table"

_metabolites, 2023, doi:10.3390/metabo13030316_

Round 1

Reviewer 1 Report

This is a well-prepared manuscript reporting a generally well-designed study (with one exception of pooling both sexes in one group, see further).

I have few minor remarks for consideration:

1) When Authors are referring to "caisson workers", the hyperbaric chamber personnel should also be included (line 32, line 41).

2) In the description of the hyperbaric exposure of subjects, there is no information on whether there was any internal attendant present (lines 118-126). 

3) The sentence with the description of the Comex Cx30 schedule (lines 123-12) with a description of air breaks suggests that air breaks were introduced by Researchers in this specific experiment "to prevent oxygen toxicity...". I suggest rewording the sentence to confirm that the original schedule was used and this original schedule includes air breaks.

4) In line 353, the Authors claim that one of the limitations is the "lack of a control group". But in this case, every subject served as each own control, as measurement was done before and after the exposure.

5) At the same time, I think that having both sexes (females and males) in the same group is a true limitation, as one can expect that there will be different responses to high oxygen exposures between them.

6) There is no indication of which values are presented on graphs (means, medians, SD, SEM, ???).

Otherwise, the manuscript is fine.

Author Response

Best regards,

On behalf of the research team,

Feiko de Jong

Reviewer 2 Report

Please see pdf file.

Author Response

(The authors gave the same response as above.)

Reviewer 3 Report

Dear Authors,

Thank you for the opportunity to familiarize yourself with the content of the manuscript. I find this article valuable. I suggest 3 fixes, namely: 1. Please clearly define the inclusion and exclusion criteria for the study 2. I have mixed feelings about the Bioethics Committee's disagreement, the study was conducted on humans 3. Some references are historical, can't they be replaced with a newer reference? best regards

Author Response

(The authors gave the same response as above.)

Reviewer 4 Report

Line 66: Consider the calculation of POT via Ran Arieli's Power Equation. This equation also includes an expression for the recovery process as well as cumulative toxicity as a function of exposure time and PO2.

Line 85: "Because oxygen exposure of a COMEX-30 treatment table 85 corresponds to 1045 UPTD". Please specify the source or calculation method to supports the UPTD value given. My calculation indicates 1206.6 UPTD.

Line 208-209  Regarding dehydration, dry mucosa, fatigue, itching was the level of hydration measured before and after exposure to the Comex 30 table? I mean either in a simple way such as urine testing (density, pH) or with  Bioimpedance analysis (BIA).

Line 259 "Seven of the ten subjects exposed to a COMEX-30 table in the present study reported symptoms of POT, such as lung discomfort and burning sensation at the end of inspiration." This high percentage of POT symptoms is not reflected in real life scenario. It could be that the diver with decompression illness treated with Comex 30 routinely receives fluid infusion and sometimes medication. In contrast, the healthy volunteer breathed, for 7.5 hours, without drinking (or at least no hydration is reported during treatment). Mixture breathing involves loss of fluids to humidify the breathed gas.

Line 286 "Because short and deep oxygen tables yield comparable therapeutic outcomes, and use of the COMEX-30 table causes mild POT in healthy subjects even without hyperoxic exposure during the previous days, the shorter tables should be used. Helium-oxygen gas mixtures may, however, play an as yet undetermined role in the future of hyperbaric medicine". Given the small number of subjects studied and some weaknesses in the study design, this conclusion is not supported by the data and needs to be either eliminated or changed to "might".

line 350. It is also true that the patient receives fluid infusion and sometimes anti-inflammatory drugs while the healthy volunteer is not reported to have drunk during the 7.5 hours of therapy. Also, in this manuscript, the recovery process  is not considered as in Ran Arieli's equation (see above).

Line 431: 26 out of 63 references (41%) are dated more than ten years ago (some are from 1970). Are they really essential for understanding the manuscript? Did the Authors of those publications not produce more recent manuscripts?

Line 473 The search for reference 21 produced no results. Please specify further.

Author Response

(The authors gave the same response as above.)
